

# Assessing the effect of fish size on species distribution model performance in southern Chilean rivers

Daniel Zamorano[1,2], Fabio A. Labra[1,3], Marcelo Villarroel[2], Shaw Lacy[4], Luca Mao[5,6], Marcelo A. Olivares[7,8] and Matías Peredo-Parada[2]

[1] Centro de Investigación e Innovación para el Cambio Climático, Facultad de Ciencias, Universidad Santo Tomás, Santiago, Chile
[2] Plataforma de Investigación en Ecohidrología y Ecohidráulica Limitada, Santiago, Chile
[3] Programa de Doctorado en Conservación y Gestión de la Biodiversidad, Facultad de Ciencias, Universidad Santo Tomás, Santiago, Chile
[4] The School for Field Studies, Center for Climate Studies, Puerto Natales, Chile
[5] Instituto de Geografía, Pontificia Universidad Católica de Chile, Santiago, Chile
[6] School of Geography, College of Science, University of Lincoln, Lincoln, United Kingdom
[7] Departamento de Ingeniería Civil, Universidad de Chile, Santiago, Chile
[8] Centro de Energía, Universidad de Chile, Santiago, Chile

Corresponding author
Matías Peredo-Parada,
matias.peredo@ecohyd.com

## ABSTRACT

Despite its theoretical relationship, the effect of body size on the performance of species distribution models (SDM) has only been assessed in a few studies, and to date, the evidence shows unclear results. In this context, Chilean fishes provide an ideal case to evaluate this relationship due to their short size (fishes between 5 cm and 40 cm) and conservation status, providing evidence for species at the lower end of the worldwide fish size distribution and representing a relevant management tool for species conservation. We assessed the effect of body size on the performance of SDM in nine Chilean river fishes, considering the number of records, performance metrics, and predictor importance. The study was developed in the Bueno and Valdivia basins of southern Chile. We used a neural network modeling algorithm, training models with a cross-validation scheme. The effect of fish size on selected metrics was assessed using linear models and beta regressions. While no relationship between fish size and the number of presences was found, our results indicate that the model specificity increases with fish size. Additionally, the predictive importance of Riparian Vegetation and Within-Channel Structures variables decreases for larger species. Our results suggest that the relationship between the grain of the dataset and the home range of the species could bias SDM, leading in our case, to overprediction of absences. We also suggest that evolutionary adaptation to low slopes among Chilean fishes increases the relevance of riparian vegetation in the SDMs of smaller species. This study provides evidence on how species size may bias SDM, which could potentially be corrected by adjusting the model grain.

## INTRODUCTION

Species distribution models (SDM) provide an important management tool to support conservation planning. SDMs generate species distribution maps that allow for more efficient and effective field inventories, suggest sites of high potential occurrence of rare species for survey planning, and permit the testing of biogeographical, ecological and evolutionary hypotheses (*Guisan & Thuiller, 2005*). Given these advantages, different international organizations (e.g., UNEP, Conservation International, IUCN, WWF) have employed SDM to address key policy objectives at a global scale (*Cayuela et al., 2009*).

Different species traits have been shown to influence species model performance (*Brotons et al., 2004*; *Segurado & Araújo, 2004*; *McPherson & Jetz, 2007*; *França & Cabral, 2016*), which could generate biases in the SDM predictions made by each species, negatively impacting its role as a management tool. One important trait is body size (*Radinger et al., 2017*), which may affect SDM performance or accuracy in different ways (*McPherson & Jetz, 2007*). Small species are often less visible and harder to capture, reducing presence/absence data availability (*McPherson & Jetz, 2007*) and implying less precise SDMs (*Stockwell & Peterson, 2002*). Additionally, positive relationships between body size, geographic range size, and home range may also affect SDM performance. Species with different home range sizes may perceive the environment of different way (*McPherson & Jetz, 2007*). As a result, model performance is expected to be higher for species whose homerange matches the climatic or environmental data used to train the SDM (*Suarez-Seoane, Osborne & Alonso, 2002*).

The relationship between body size, geographic range size, and home range in fish species may also affect SDM variable selection. For example, larger fishes can disperse farther than small fishes and are expected to be more significantly restricted by in stream barriers (*Radinger & Wolter, 2015*; *Radinger et al., 2017*). Conversely, the lower dispersal ability of smaller fishes implies a smaller response to anthropogenic drivers (*Radinger & Wolter, 2015*; *Radinger et al., 2017*). Thus, body size can potentially affect SDMs by influencing predictor variable selection.

To date, the effect of body size on distribution models has been tested in different taxa with unclear results (e.g., *McPherson & Jetz, 2007*; *França & Cabral, 2016*; *Morán-Ordóñez et al., 2017*; *Radinger et al., 2017*). In the case of fish, this relationship has been tested indirectly for riverine fish (*Radinger et al., 2017*) and directly for marine and estuary fishes (*Perry et al., 2005*; *França & Cabral, 2016*). For example, *Radinger et al. (2017)* tested future distributions according the body size of river fishes and showed that smaller-body fishes are less sensitive to anthropogenic intervention in the river network due to their smaller home ranges.

The native ichthyofauna in Chile comprises a total of 44 species, including two lampreys (*Habit, Dyer & Vila, 2006*) and is characterized as being highly endemic, adapted to low slope rivers, and having small body sizes (*Vila, Fuentes & Contreras, 1999*; *Vila et al., 2006*; *Habit, Dyer & Vila, 2006*). In addition to its high biogeographic value, the Chilean ichthyofauna is broadly endangered, with only two species (*Cheirodon austral* Eigenmann, 1927 and *Mugil cephalus* Linnaeus, 1758) currently classified as non endangered, which

provides additional modeling challenges. For example, all species have scant distribution data due to the type of research conducted in the area, and adult body size ranges from 5 to 40 cm, making them additionally difficult to capture and sample. Given these challenges, assessing how fish size affects SDMs in Chile will allow us to determine better model methodologies suited to the region's small-sized, data-deficient and endangered species.

Our aim is to evaluate the relationship between fish size and SDM goodness-of-fit using three approaches: (1) assess the relationship between fish size and data availability, (2) assess the relationship between fish size and model performance, and (3) compare predictor variable participation and patterns according to fish size. We focus on two well-studied southern Chilean river basins, Bueno and Valdivia, and model nine native species.

# METHODS

## Study area and modeled species

The study area covers the Valdivia and the Bueno River basins located in the southern zone of Chile between 39.33° and 41.08°S (Fig. 1). The Valdivia River basin has a pluvial hydrological regime and is characterized by a chain of interconnected lakes at higher altitudes. The upper section of the Bueno River basin has a pluvio-nival regime, while the middle and lower parts of the basin are governed by a pluvial regime (*Errázuriz et al., 1998*).

Our study examined nine native freshwater fish species (Table 1): *Aplochiton taeniatus* Jenyns, 1842, *Aplochiton zebra* Jenyns, 1842, *Basilichthys microlepidotus* (Jenyns, 1841), *Brachygalaxias bullocki* (Regan, 1908), *Cheirodon australe, Diplomystes camposensis* (*Arratia, 1987*), *Galaxias maculatus* (Jenyns, 1842), *Percilia gillissi* Girard, 1855, and *Trichomycterus areolatus* (Valenciennes in Cuvier & Valenciennes, 1840). Statistical analyses of the effect of body size were carried out using theoretical species maximum length, which is available for all these species. Maximum length estimates were obtained from official species descriptions provided the conservation assessment of each species, developed by the Chilean Ministry of the Environment, with the only exceptions being *B. bullocki* and *B. australis* (Table 1).

## Modeling methods
### Model grain

To build the SDM database, the drainage network of the study area was divided into segments. We considered river segments as having homogeneous hydromorphological conditions with no significant confluences and 2 and 10 km in length. Each segment represented an analysis unit in which the presence and SDM predictor variables for each species were evaluated. This segment definition was generated using cartographic information, visual interpretation of Google Earth imagery (https://www.google.com/intl/es/earth/), and Arc GIS version 9.2 (http://desktop.arcgis.com/es/; Esri, Redlands, CA, USA). To characterize a set of hydrological variables for the study area, we used the national official drainage network generated by the Military Geographic Institute (Instituto GeográficoMilitar, Government of Chile).

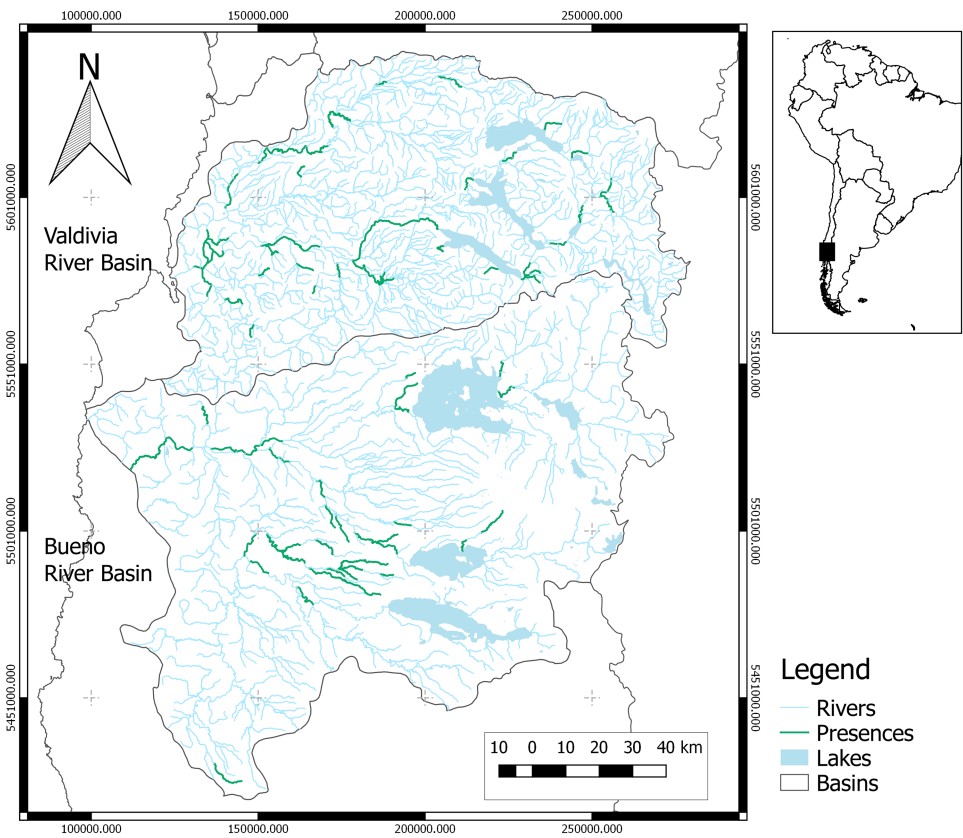

**Figure 1** **Study area.** Map shows study basins (Valdivia and Bueno). Green river reaches contain species records. The black square in the map of South America shows the study area.

### Species occurrence data

Historical records of the georeferenced presences of the study species were obtained from the Ministry of the Environment's (Ministerio del Medio Ambiente, Government of Chile) database on freshwater organisms. This database was generated by collecting published databases of scientific samples in the study area (*Ministerio de Energía División de Desarrollo Sustentable, 2016*). Since this biogeographic region contains 16 native fish species (*Habit, Dyer & Vila, 2006*), our study accounts for 56% of extant species in this region of Chile (*Vila et al., 2006*). The remaining species had insufficient numbers of occurrence records inthe database (between 0 and 10) and were not considered.

A field sampling campaign was conducted in the study area to increase the records by specie presents in the government data. The sampling was performed in December 2015 and January 2016 using electrofishing equipment (SAMUS, model 745G). We collected all fish along a 100 m section of river, with sampling times of 45 to 60 min, depending on the hydromorphological features of the site. Seven rivers were sampling: Llancahue River, Pinchichirre River, Nilfe Channel, Quinchilca River, Los Nadis River, Punahue Channel and Cahuinalhue River, recording nine presences to seven species. All collected fish were

Zamorano et al. (2019), *PeerJ*, DOI 10.7717/peerj.7771

**Table 1  Model summaries.** Columns on order: Modeled species, natural log of maximum body size in cm, conservation category assigned by the Chilean government, number of recorded presences, K folds used in the model, number of modeled presences, and TSS, AUC, Balanced accuracy (B Acc), Sensibility (Sens) and Specificity (Spec) values for each model with their standard errors (SE).

| Species | Ln size | Conservation category | Presences | K folds by model | Presences used on model | TSS | TSS SE | AUC | AUC SE | B Acc | B Acc SE | Sens | Sens SE | Spec | Spec SE |
|---|---|---|---|---|---|---|---|---|---|---|---|---|---|---|---|
| *Aplochiton taeniatus*[a] | 3.40 | EN | 28 | 3 | 17 | 0.78 | 0.03 | 0.92 | 0.03 | 0.89 | 0.02 | 0.83 | 0.06 | 0.94 | 0.03 |
| *Aplochiton zebra*[b] | 3.33 | EN | 22 | 3 | 16 | 0.47 | 0.05 | 0.87 | 0.02 | 0.73 | 0.03 | 0.57 | 0.02 | 0.90 | 0.03 |
| *Basilichthys australis*[c] | 3.60 | FP | 56 | 3 | 23 | 0.83 | 0.08 | 0.98 | 0.01 | 0.92 | 0.04 | 0.86 | 0.08 | 0.98 | 0.01 |
| *Brachygalaxias bullocki*[d] | 1.70 | VU | 37 | 3 | 28 | 0.45 | 0.05 | 0.84 | 0.01 | 0.72 | 0.03 | 0.64 | 0.08 | 0.80 | 0.03 |
| *Cheirodon australe*[e] | 1.95 | VU | 56 | 3 | 21 | 0.74 | 0.04 | 0.94 | 0.01 | 0.87 | 0.02 | 0.81 | 0.05 | 0.93 | 0.02 |
| *Diplomystes camposensis*[f] | 3.21 | EN | 34 | 2 | 14 | 0.68 | 0.18 | 0.90 | 0.07 | 0.84 | 0.09 | 0.79 | 0.15 | 0.89 | 0.03 |
| *Galaxias maculatus*[g] | 2.77 | FP | 30 | 3 | 17 | 0.55 | 0.01 | 0.92 | 0.00 | 0.77 | 0.00 | 0.76 | 0.07 | 0.79 | 0.07 |
| *Percilia gillissi*[h] | 2.20 | EN | 62 | 3 | 33 | 0.55 | 0.06 | 0.87 | 0.03 | 0.77 | 0.03 | 0.79 | 0.08 | 0.76 | 0.06 |
| *Trichomycterus areolatus*[i] | 2.71 | VU | 72 | 3 | 36 | 0.46 | 0.01 | 0.82 | 0.01 | 0.73 | 0.01 | 0.64 | 0.03 | 0.82 | 0.03 |

**Notes.**

Reference to fish size:

[a]*Ministerio del Medio Ambiente (2011a)*.

[b]*Ministerio del Medio Ambiente (2011b)*.

[c]*Cifuentes et al. (2012)*.

[d]*Ministerio del Medio Ambiente (2008d)*.

[e]*Ministerio del Medio Ambiente (2008a)*.

[f]*Ministerio del Medio Ambiente (2008e)*.

[g]*Ministerio del Medio Ambiente (2008b)*.

[h]*Froese & Pauly (2017)*.

[i]*Ministerio del Medio Ambiente (2008c)*.

identified to species level. The electrofishing was approved by the National Fisheries Service, permit number 514.

Each presence record was associated with the closest river segment in the GIS, thus building a presence database for species distribution modeling. Overall, 118 river segments had at least one presence record. The total number of presences for each species across the two study basins ranged between 22 and 72, but the modeling only considered one record per segment, so modeled presences ranged between 14 and 36 (Table 1).

### Predictor variables

The predictor variables or features considered were accumulated rainfall, catchment area, source-of-flow, altitude, slope, channel width, percent riparian vegetation, land-use, cross-channel structures, and within-channel structures (Table 2).

Annual rainfall was obtainedby relating the isolines of average annual rainfall (Dirección General de Aguas, Government of Chile) over the segment basin. The catchment area was calculated using a 1 km $\times$1 km DEM image (Landsat 7 images from 2015, https://landsat.usgs.gov/) using the Hydrology package in ArcGIS for each segment basin considering the accumulated catchment area. Source-of-flow, that represents the geographical origin of the flow, was obtained and adapted from the published REC-Chile classification (*Peredo-Parada et al., 2011*), and each category was used independently as a Boolean variable. Final source-of-flow variables were lakes, floodplains, valleys and mountains. Altitude and slope were estimated using the altitudes atthe downstream end of each river segment based on the DEM. Channel width, riparian vegetation cover, land-use, cross-channel structures, and within-channel structures were estimated using a visual analysis of Google Earth imagery. The channel width was calculated as the mean of three points along the reach. Riparian vegetation coverage was considered up to 50 m from the stream, with sections and land-use classes defined based on evaluation up to 200 m. Land use was defined in three categories: Antr-Antr = on both banks over 50% of the area has anthropogenic interventions; Nat-Antr = on one bank over 50% of the area has anthropogenic interventions; and Nat-Nat = on both banks less than 50% of the area has anthropogenic interventions. Within-channel structures included roads parallel to the river, bank reinforcement, river channel maintenance structures, and channelization, among others. Cross-channel structures included bridges, dams, and water-intake structures.

## Model training and evaluation

We used neural network (NNET) algorithms to estimate SDMs (*Stern, 1996*). This method was chosen based on their good performance with presence and absence or pseudoabsencesfor species-distribution data (*Mastrorillo et al., 1997*; *Elith & Leathwick, 2009*). NNET is derived from a simple model that mimics the structure and function of the brain and maximizes the prediction during the model-training phase by comparing actual outputs with desired outputs (*Manel, Dias & Ormerod, 1999*). All analyses were performed in R (v 3.5.0) using the Caret package (*Kuhn, 2008*).

Models were trained using a 3- or 2-fold cross-validation scheme, according to records by species (Table 1). Pseudoabsences were set at twice the number of observed species

Zamorano et al. (2019), *PeerJ*, DOI 10.7717/peerj.7771

**Table 2  Results of analyses of different response variables as a function of fish size.** Columns in order: response variables, *p*-value of the Phi coefficient, *p*-value of the RM test to residuals, evaluation of the presence of a significant covariance (True or False), pseudo $R^2$ (with link transformation of response variable), mean estimate of analysis, Std. Error of analysis, and the *Z*-value and *P*-value of the beta regression analysis.

| Response variables | Phi coefficient *p*-value | RM test *p* value | Significant covariance | Pseudo $R^2$ | Mean estimate | Std. error | *Z* value | *P* value | Significance |
|---|---|---|---|---|---|---|---|---|---|
| Balanced accuracy | 0.03 | 0.68 | F | 0.26 | 0.31 | 0.21 | 1.51 | 0.132 | |
| Sensitivity | 0.03 | 0.70 | F | 0.10 | 0.16 | 0.23 | 0.70 | 0.484 | |
| Specificity | 0.03 | 0.40 | F | 0.41 | 0.56 | 0.28 | 2.04 | 0.042 | ** |
| TSS | 0.03 | 0.68 | F | 0.26 | 0.31 | 0.21 | 1.51 | 0.132 | |
| AUC | 0.03 | 0.48 | F | 0.48 | 0.36 | 0.23 | 1.57 | 0.117 | |
| Cross-channel structure | 0.03 | 0.19 | F | 0.11 | −0.21 | 0.25 | −0.82 | 0.411 | |
| Within-channel structure | 0.03 | 0.47 | F | 0.54 | −0.65 | 0.21 | −3.15 | 0.002 | *** |
| Land use: Antr-Antr | 0.02 | 0.38 | F | 0.21 | −0.13 | 0.54 | −0.24 | 0.813 | |
| Land use: Nat-Antr | 0.03 | 0.20 | F | 0.39 | −0.52 | 0.49 | −1.07 | 0.283 | |
| Land use: Nat-Nat | 0.03 | 0.27 | F | 0.31 | −0.26 | 0.29 | −0.92 | 0.358 | |
| Altitude | 0.02 | 0.29 | F | 0.12 | −0.41 | 0.38 | −1.09 | 0.277 | |
| Slope | 0.02 | 0.36 | F | 0.04 | −0.21 | 0.42 | −0.50 | 0.616 | |
| Channel width | 0.02 | 0.99 | F | 0.15 | 0.36 | 0.44 | 0.80 | 0.423 | |
| Riparian vegetation | 0.03 | 0.49 | F | 0.49 | −0.50 | 0.22 | −2.27 | 0.023 | ** |
| Catchment area | 0.02 | 0.61 | F | 0.15 | −0.51 | 0.60 | −0.85 | 0.397 | |
| Annual rainfall | 0.02 | 0.49 | T | 0.47 | 0.31 | 0.46 | 0.66 | 0.508 | |
| Source-of-flow: Lakes | 0.03 | 0.68 | F | 0.25 | −0.47 | 0.33 | −1.41 | 0.158 | |
| Source-of-flow: Foodplain | 0.03 | 0.30 | F | 0.03 | 0.19 | 0.44 | 0.44 | 0.659 | |
| Source-of-flow: Mountains | 0.03 | 0.01 | F | 0.26 | −0.27 | 0.33 | −0.81 | 0.420 | |
| Source-of-flow: Valleys | 0.03 | 0.74 | F | 0.25 | −0.53 | 0.33 | −1.62 | 0.106 | |

presences and were randomly selected. We designated as the final model the ensemble average of each k-fold model by its weighting each individual model by their Area Under the Curve statistic (AUC) of the Receiver Operating Characteristic (ROC).

The Caret R package was used to fit NNET models and tune the model's two hyperparameters, namely, the weight decay for successive neural layers ("*decay*") and the number of hidden units ("*size*"). The grid search procedure examined weight decay values ranging between 4 and 6, while the number of hidden units was allowed to vary between 0.05 and 0.9. Both hyperparameter ranges were calibrated by a trial and error process, optimizing the model performance.

Occurrence probabilities were categorized as presence/absence for all models. Thresholds were determined to maximize the sum of sensitivity and specificity (MaxSens+Spec; PresenceAbsence package in R v 3.5.0) (*R Core Team, 2017*). This criterion is independent of the theoretical prevalence (*Manel, Dias & Ormerod, 1999*; *Allouche, Tsoar & Kadmon, 2006*), causing the distribution of rare species to be overpredicted. In our case, the theoretical prevalence in the study area for all the species is close to 0.5, but the presence of our study species is low, requiring a relaxation of this criterion when defining the threshold that allows for the definition of each of the species distribution across the studied watersheds. All the presences and environmental data by river segment in supplemental material (Appendix S1).

## Relationship between fish size and models

As proxy of body size, $\log_{10}$—transformations of maximum length (*max. length*) were calculated for each species. We assessed the relationship between *max. length* and the number of historical records in both basins, with balanced accuracy, sensitivity, specificity, true skill statistic (TSS), and area under curve (AUC), all different performance metrics (*Altman & Bland, 1994*; *Allouche, Tsoar & Kadmon, 2006*; *Velez et al., 2007*; *Kuhn, 2008*), and with variable importance to each predictor variable. Sensitivity evaluates the proportion of actual presences that are correctly classified and specificity evaluates the proportion of actual negatives that are correctly identified (*Allouche, Tsoar & Kadmon, 2006*). AUC, TSS and balanced accuracy evaluate overall performance using different approximations, being valuable contrast them all (*Velez et al., 2007*; *Kuhn, 2008*). In all this cases, obtain a value of 1 to this metrics represent the best performance, and in the AUC case, 0.5 represent the worst performance. Variable importance is calculated with "Weights" method (*Gevrey, Dimopoulos & Lek, 2003*), and the highest importance is 100.

The statistical test of the relationship between *max. length* and historical records was a classical linear model (*lm*) (*Chambers, 1992*), while the other comparisons were evaluated with beta regression analysis (*betareg*) (*Cribari-Neto & Zeileis, 2010*). This test assumes response variables to be in the standard unit interval (0, 1) and to be beta distributed, and it is more precise in performance metrics cases or variable importance cases where the values are at a fixed interval. Rescaled Moment test (RMtest) and Q–Q plots were applied to all residual models to evaluate model validity (*Das & Imon, 2016*; *Kozak & Piepho, 2018*). While, the normality test is used on observed data to evaluate parametricity, in regressions the true errors are unobserved, being a common practice to use residuals as substitutes for

observed data in tests for normality (*Das & Imon, 2016*). In this context, we used RMtest to evaluate normality of residuals and validate *betareg* and *lm* models. RM test correct shrinkage and superimposed normality effect of the residual data, improve its normality evaluation (*Imon, 2003*; *Rana, Midi & Imon, 2009*).

To avoid a potential bias from conservation status by specie on all statistical tests, we introduced this category as a covariable. Given the hierarchical relationship between conservation status categories, we transformed this to a numeric covariable, with Least Concern (LC) being 1, Vulnerable (VU) as 2, and Endangered (EN) as 3. Finally, an *a priori betareg* test was developed to discard the relationship between the number of presences by model and metric performance models, allowing for directly linking fish size with model indicators.

## RESULTS

All models showed good performance, with ROC AUC varying between 0.82 and 0.98, while TSS varied between 0.45 and 0.83. The best trained model was for *B. australis* (AUC = 0.98 and TSS = 0.83), while the worst trained model was for *T. areolatus* (AUC = 0.82 and TSS = 0.46) (Table 1). The number of presences used by the model did not show a significant relationship with model performance metrics (*betareg* to AUC, *p* value = 0.104, *betareg* to TSS, *p* value = 0.159).

Channel width and Catchment area were the most important variables in most species models, excluding *A. taeniatus*, *G. maculatus*, and *P. gillissi* models, where Annual rainfall, Land-use: Antr-Antr, and Altitude were the most important, respectively. However, if we added the Source-of-Flow participation by category, its participation was more relevant than the Channel width and Catchment area for all species (Table 2).

### Historical records

The number of presences and conservation status did not correlate with fish size (*lm* test: Conservation status, coefficient = −4.7, std. error = 7.9, *t* value = −0.56, *p* value = 0.57; Historical records, coefficient = −8.1, std. error = 9.67, *t* value = -0.84, *p* value = 0.43). Model residuals show a normal distribution (Chisq = 1.83, *p* value = 0.59). The species with the lowest presence was *A. zebra*, with 22 records and 28 cm maximum length and is one of the largest native species found in the basins. The species with most historical records was *T. areolatus,* with 72 presences and a maximum length of 15 cm. However, many of the *T. areolatus* presences were spatially clustered, and the model was trained using only 36 presence records (Table 1).

### Metric performance

Of all the evaluated performance metrics, only specificity showed a significant positive relationship with fish size (Fig. 2). In most modeled species, we observed residuals being normally distributed considering Q–Q plots and RMtest (Tables 3A and 3B). The only one exception is Source-of-flow: Mountains, with a non significant RMtest. Specificity had a positive relationship with fish size (Fig. 2). Other metrics did not show significant relationships with fish size (Table 3).

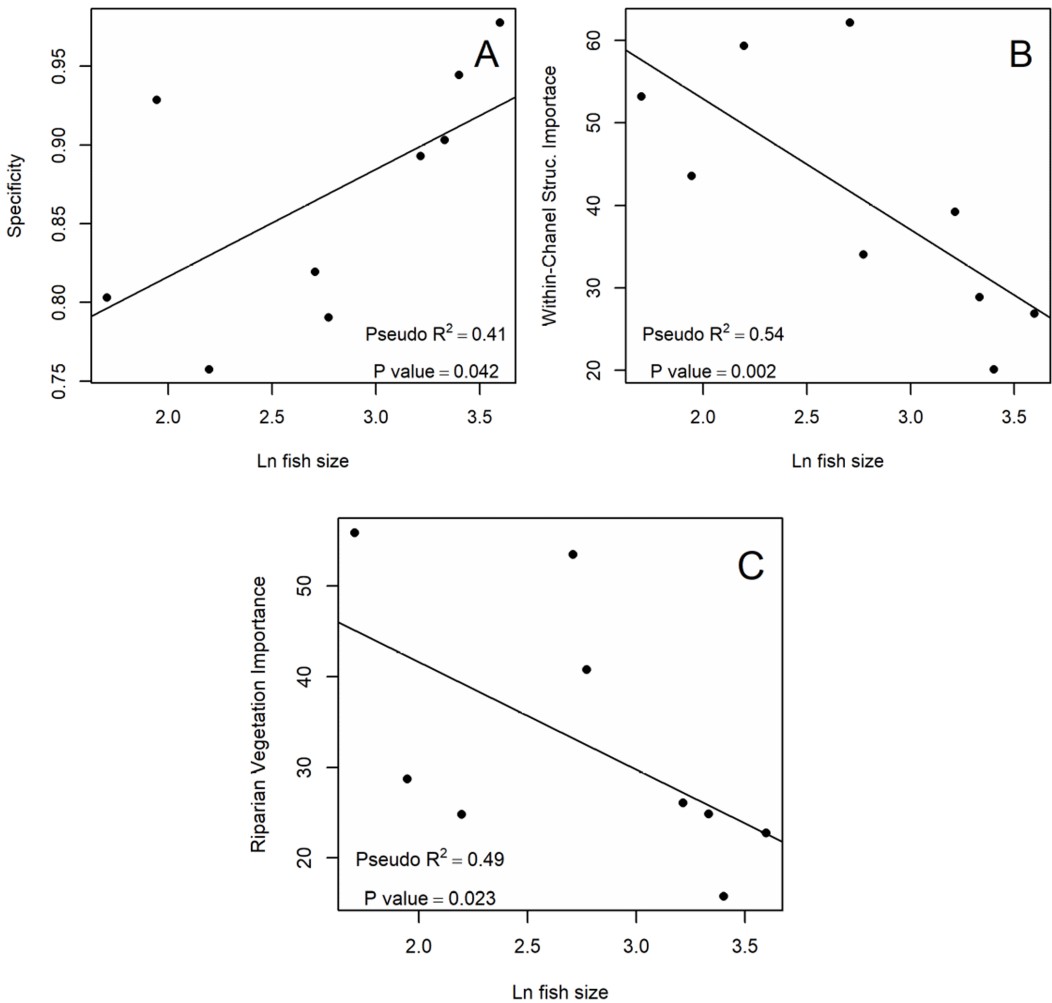

**Figure 2  Significant relationships between SDM metrics and (Ln) fish size.** (A) Positive relationship between the specificity of the SDM and Ln fish size. (B) Negative relationship between importance in the SDM of Within-Channel Structures and Ln fish size. (C) Negative relationship between importance in the SDM of Riparian Vegetation and Ln fish size. Text in each plot shows pseudo $R^2$ and *p*-value obtain with beta regression analysis.

## Variable importance

When we examined the significance of individual predictor variables, we observed that only Riparian Vegetation and Within-Channel Structures showed a significant relationship with fish size. Again, we observed residuals being normally distributed, thus validating the statistical analysis (Tables 3A and 3B). In these cases, the fitted models for larger fish tended to give less importance to these variables than models for small fish, showing a negative relationship (Fig. 2). Other variables did not show significant relationships with fish size (Tables 3A and 3B).

When exploring the relationship between occurrence probability by SDM and percentage of Riparian Vegetation and number of Within-Channel Structures (Fig. 3), we found no clear patterns. In the Riparian Vegetation case, smaller species tended to increase occurrence

Zamorano et al. (2019), *PeerJ*, DOI 10.7717/peerj.7771

**Table 3A  Variable participation by modeled species.** Bold values indicate the largest values by species. Final row represents the participation mean by variable.

| Species | Within-channel structures | Cross-channel structures | Land use: Antr-Antr | Land use: Nat-Antr | Land use: Nat-Nat | *Land use: All* | Altitude | Slope | Channel width |
|---|---|---|---|---|---|---|---|---|---|
| *Aplochiton taeniatus* | 25.40 | 20.11 | 10.63 | 2.73 | 13.60 | 26.96 | 48.39 | 17.09 | 58.79 |
| *Aplochiton zebra* | 32.84 | 28.87 | 13.85 | 5.12 | 16.49 | 35.47 | 14.14 | 66.58 | **95.34** |
| *Basilichthys australis* | 16.08 | 26.85 | 6.01 | 3.84 | 3.60 | 13.45 | 37.40 | 9.62 | 67.91 |
| *Brachygalaxias bullocki* | 35.48 | 53.20 | 23.58 | 27.80 | 29.32 | 80.70 | 24.02 | 34.48 | 50.85 |
| *Cheirodon australe* | 20.30 | 43.59 | 14.01 | 11.19 | 15.57 | 40.76 | 63.42 | 51.03 | 57.85 |
| *Diplomystes camposensis* | 19.88 | 39.20 | 28.79 | 13.95 | 34.79 | 77.53 | 34.61 | 32.04 | 75.77 |
| *Galaxias maculatus* | 57.51 | 34.04 | **97.22** | 87.72 | 13.65 | **198.59** | 59.40 | 68.30 | 35.27 |
| *Percilia gillissi* | 33.63 | 59.36 | 6.00 | 12.40 | 8.67 | 27.07 | **81.60** | 11.84 | 79.45 |
| *Trichomycterus areolatus* | 26.79 | 62.18 | 20.75 | 24.51 | 15.50 | 60.76 | 47.99 | 39.15 | **99.26** |
| Summary participation | 26.79 | 40.82 | 24.54 | 21.03 | 16.80 | 62.37 | 45.66 | 36.68 | **68.94** |

Zamorano et al. (2019), *PeerJ*, DOI 10.7717/peerj.7771

**Table 3B  Variable participation by modeled species.** Bold values indicate the largest values by species. Final row represents the participation mean by variable.

| sp | Riparian vegetation | Catchment area | Annual rainfall | Source-of-flow: Lakes | Source-of-flow: Plain | Source-of-flow: Mountains | Source-of-flow: Valleys | *Source-of-flow: All* |
|---|---|---|---|---|---|---|---|---|
| *Aplochiton taeniatus* | 15.77 | 30.88 | **100.00** | 9.39 | 14.93 | 28.50 | 37.39 | 90.21 |
| *Aplochiton zebra* | 24.83 | 18.12 | 77.49 | 20.14 | 21.10 | 29.33 | 25.62 | 96.19 |
| *Basilichthys australis* | 22.75 | **94.50** | 8.12 | 11.10 | 8.31 | 2.94 | 4.78 | 27.13 |
| *Brachygalaxias bullocki* | 55.88 | **99.38** | 40.38 | 30.21 | 33.43 | 16.29 | 27.18 | 107.11 |
| *Cheirodon australe* | 28.69 | **100.00** | 26.50 | 24.72 | 1.18 | 30.24 | 43.95 | 100.09 |
| *Diplomystes camposensis* | 26.05 | **100.00** | 31.62 | 30.51 | 15.45 | 37.68 | 6.94 | 90.57 |
| *Galaxias maculatus* | 40.79 | 31.67 | 33.86 | 62.37 | 23.40 | 51.51 | 30.35 | 167.63 |
| *Percilia gillissi* | 24.77 | 39.48 | 75.92 | 43.56 | 7.01 | 42.01 | 19.98 | 112.56 |
| *Trichomycterus areolatus* | 53.47 | 59.72 | 34.26 | 16.63 | 54.65 | 31.11 | 39.94 | 142.33 |
| Summary participation | 32.56 | **63.75** | 47.57 | 27.63 | 19.94 | 29.96 | 26.24 | **103.76** |

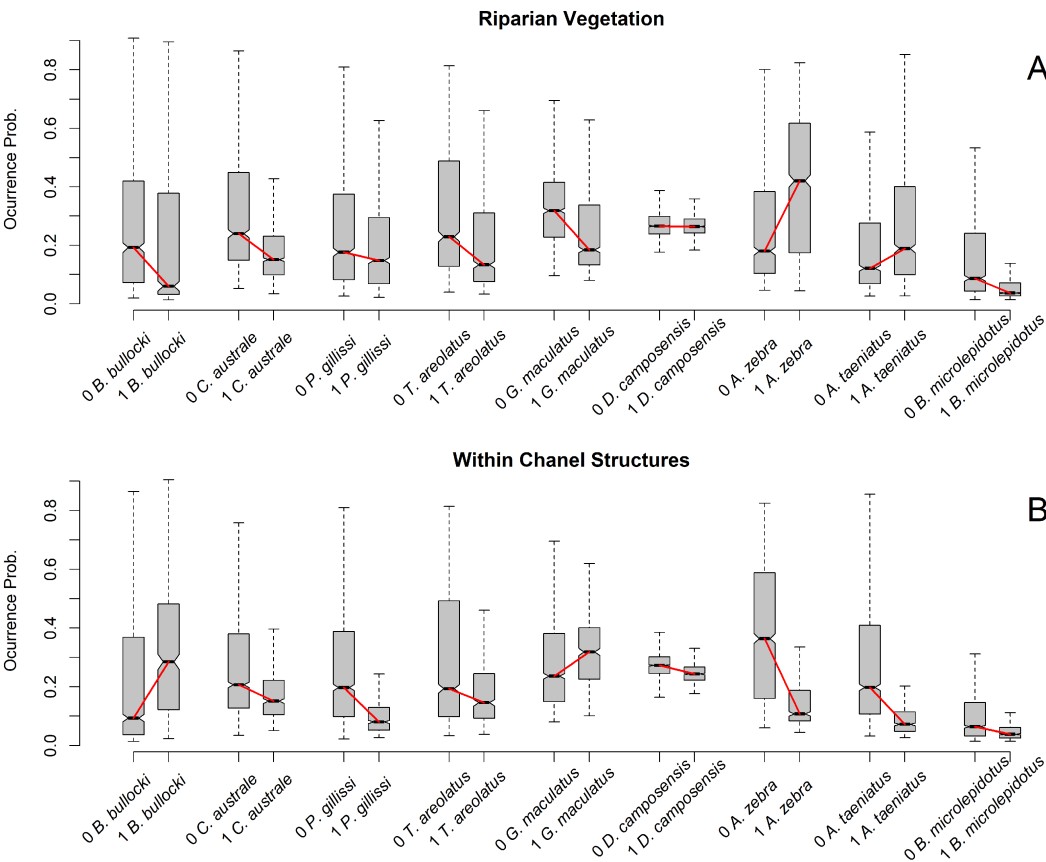

**Figure 3** **Occurrence probability boxplot according to species and predictor variable class.** (A) Reaches were classified as 1 = 100% riparian vegetation and 0 < 100% riparian vegetation. (B) Reaches were classified as 1 = one or more within-channel structures and 0 = no within-channel structures. Boxplots show the occurrence probability of each species model by reach class. Species on the *X* axis are sorted by size, with larger species on the right. Boxplot is represent by five values:the extreme of the lower whisker is the lowest datum still within 1.5 Interquartile range, the lower box extreme is the first quartile,the middle line is the median, the upper box extreme is the third quartile andthe extreme of the upper whisker is the highest datum still within 1.5 Interquartile range. Outliers are excluded to clarify visualization.

probability on reaches with less riparian vegetation, but the pattern of larger species was unclear. In contrast, larger species tended to show higher occurrence probabilities on reaches without Within-Channel Structures, while there was no clear pattern in smaller fish species.

## DISCUSSION

The principal result in the current study showed that SDM of smaller fishes has lower specificity, tending to overpredict presences. This finding has important implications for conservation, management and the science related to SDMs, especially in countries as Chile, with particularly small species. While, it is not recommended modeling a group of species with the same methodology, is very common in papers and management projects. In this context, our results show the relevance of evaluate potential bias according to physiological

and ecological traits by specie, identifying species that need different methodologies to make more accurate model.

The good performance obtained for all the trained models shows how the hyperparametrization process using software such as Caret allows us to obtain good SDMs (*Kuhn, 2008*). Additionally, good SDM performance using few presences may also occur when models are projected on regions that have highly similar environmental conditions to where the species occur (*Pearson et al., 2006*).

Regarding predictor variable importance, the most important variables were those that represented hydrological regime (Source-of-flow) and flow discharge (Catchment area or Accumulated rainfall), with both variables representing segment scale (*Frissell et al., 1986*; *Snelder & Biggs, 2002*). Large-scale predictor variables have more participation across models, and as the geographical scale of the variables decreases, their participation in the model does, as well; in addition, their importance is resolved species by species. This notion corresponds with the hierarchical framework of stream habitat proposed in the literature (*Frissell et al., 1986*; *Snelder & Biggs, 2002*; *Creque, Rutherford & Zorn, 2005*; *Steen et al., 2008*; *Peredo-Parada et al., 2011*).

## Records by fish species

Although many authors mention the relationship between body size and data availability (*Boone & Krohn, 1999*; *McPherson & Jetz, 2007*; *França & Cabral, 2016*), we found no studies in which this relationship was statistically evaluated. The premise that larger fish species are more detectable than smaller fish species depends on the sampling method and species characteristics (diurnal or nocturnal, color, habitat selection, etc.) (*Boone & Krohn, 1999*). Thus, a detection bias may occur if sampling is conducted at the community or ensemble level using a single sampling methodology. However, if sampling is stratified or designed for a given species population using adequate methodologies, we would expect this bias to be corrected. In this study, presence information was obtained from a governmental database generated from published scientific studies, with sampling methodologies having been determined by studies focused on independently assessing species abundances and distributions. This situation explains the lack of relationship between fish size and presence records in our study.

## Fish size and model fit

Our results regarding the body-size effects on SDM performanceare relevant to the unresolved debate about this expected theoretical relationship. While *Morán-Ordóñez et al. (2017)* found no significant relationship between body size and model performance for trees and birds, *França & Cabral (2016)* successfully related model performance to species feeding mode and estuarine functional groups, with body size having a marginally significant contribution to model performance. In studies aimed at river fish, while *Radinger et al. (2017)* and *Filipe, Cowx & Collares-Pereira (2002)* found that fish size increased model performance, *Markovic, Freyhof & Wolter (2012)* did not observe this pattern.

Our results show that the specificity performance metric increases with fish size, suggesting that the ability to predict true absences increases with fish size. This outcome

means that models for smaller fishes tend to overpredict presences. These results reflect those of *Lobo & Tognelli (2011)*, who found using virtual species, models with randomly determined presences tend to have lower specificity relative to models with spatially biased presences. We suggest that in our study, smaller fish may have a statistically random distribution as a result of a potential mismatch between the study grain and their home range. This outcome would be the consequence of a random distribution of predictor variables used to characterize fish habitat (*McPherson & Jetz, 2007*).

This finding has important implications for conservation and management. First, it suggests that the environmental resolution used in the evaluation should be commensurate with the modeled species' home range. However, when this outcome is not possible (due to financial or logistic resources available, for example), then explicit assessment is necessary to detect any potential biases in model performance for species of different body sizes. In this way, it is also important to assess both metrics that represent general performance, such as AUC, TSS or Accuracy, and to include other specific metrics that may provide relevant information, such as any likelihood of overprediction of smaller species.

Our use of theoretical maximum length as the best available estimate of species body size does not necessarily reflect actual maximum body size in the study areas, and certainly may result in biases or estimation errors. Explicit assessment of possible biases stemming from the use of theoretical or literature based maximum lengths versus empirical estimates of maximum body length would certainly improve our current understanding of the interaction between body size and SDM model performance. This could be achieved by explicitly describing observed median and maximum body size of the set of recorded presences within study hydrological basins, allowing body size to reflect current ecological factors modifying this trait. Such a study design would allow the assessment of either fine scale habitat alterations, or possible impacts of invasive species abundance. It must be noted that such a study is not free of logistic and bioethical constraints. Most important, most of the fish species addressed in this study face important threats, and wide scale sampling and individual monitoring may pose additional stress to these individuals. As a result, an important source of data for such a study could be the samples collected in the development of environmental impact assessment (EIA) studies. However, under Chilean legislation, EIA sampling designs respond the project developer specific goals, and body size sampling is not often included among the variables registered.

## Participation by predictor variable

The predictor variables that show trends in model importance according to fish size are both obtained at smaller scales (reach scale) (*Frissell et al., 1986*). The participation of riparian Vegetation and Within-Channel Structures decreases with fish size, suggesting that these variables are not relevant to estimating the ecological niche of large fish species. For Within-Channel Structures, *Radinger et al. (2017)* found that these may cause larger habitat reduction for larger fish species by creating stream barriers that limit their dispersal. This finding is consistent with our results, which show how the largest three species have low occurrence probabilities on reaches with almost one structure detected.

The relationship between riparian vegetation cover and fish size may be determined by the association between riparian vegetation and river slope. Rivers with higher slopes have short alluvial plains that allow for more riparian vegetation (*Stefunkova, Neruda & Vasekova, 2019*). However, the morphologies of Chilean fish species are not adapted to high-slope habitats (*Arratia, 1987*; *Arratia, 1992*); therefore, we presume that riparian vegetation acts as a proxy of habitat conditions adequate for smaller species. Functionally, this notion means that the occurrence probability of smaller species increases in rivers with less riparian vegetation cover.

The current study presumed that no interspecific effects occurred between species that impacted SDMs, particularly the potential role of the invasive species. Invasive species have been shown to influence native fish body size (*Blanchet et al., 2010*), which can have significant effects on SDM performance. In Chile, invasive species such as rainbow trout (*Oncorhynchus mykiss*) and brown trout (*Salmo trutta*) have strong impacts on fishes (*Pardo, Vila & Capella, 2009*; *Arismendi et al., 2014*) and may benefit from riparian vegetation conservation, especially in upland areas (*Lacy, Ugalde & Mao, 2018*). Regrettably, in the current study, we were unable to evaluate the effect of invasive species on body size. Future studies should seek to integrate ecological effects, especially effects caused by invasive species, into the development of SDMs.

Finally, we want to highlight the use of source-of-flow as a predictor variable in our study; we have not found this use in prior researchon river-species modeling, and it is especially important in torrential river systems, such as those found in Chile. These systems have short runs, with relatively large lakes, glaciers, or salt pans that significantly affect hydrological and hydraulic conditions. The source-of-flow variable is implemented in river evaluations in New Zealand (*Snelder & Biggs, 2002*) and Chile (*Peredo-Parada et al., 2011*), which would facilitate its use in SDMs.

## CONCLUSIONS

In this study, we found relationships between fish size and model performance, increasing specificity along with fish size. This is new evidence in support of this classical theoretical relationship, supporting the idea that model performance is affected by species characteristics. We also show how predictor variables have different importance in the models, according to scale, with Source-of-Flow, Catchment area or Accumulated rainfall being relevant to all of the models, and Riparian Vegetation and Within-Channel Structures being relevant variables according to the ecology of the species. Further investigations should consider this potential source of bias to determine management and conservation objectives from SDMs by either modifying the methodology or conducting *a posteriori* evaluations.

### Funding

The authors received no funding for this work.

## Competing Interests

Daniel Zamorano and Marcelo Villarroel were employed by Plataforma de Investigación en Ecohidrología y Ecohidráulica Limitada. Matías Peredo is director in chief of Plataforma de Investigación en Ecohidrología y Ecohidráulica Limitada. The other authors declare there are no competing interests.

## Author Contributions

- Daniel Zamorano conceived and designed the experiments, performed the experiments, analyzed the data, prepared figures and/or tables, approved the final draft.
- Fabio A. Labra conceived and designed the experiments, analyzed the data, prepared figures and/or tables, authored or reviewed drafts of the paper, approved the final draft.
- Marcelo Villarroel performed the experiments, contributed reagents/materials/analysis tools, approved the final draft.
- Shaw Lacy performed the experiments, authored or reviewed drafts of the paper, approved the final draft.
- Luca Mao analyzed the data, authored or reviewed drafts of the paper, approved the final draft.
- Marcelo A. Olivares authored or reviewed drafts of the paper, approved the final draft.
- Matías Peredo-Parada conceived and designed the experiments, performed the experiments, analyzed the data, prepared figures and/or tables, authored or reviewed drafts of the paper, approved the final draft.

## Field Study Permissions

The following information was supplied relating to field study approvals (i.e., approving body and any reference numbers):

The electrofishing was approved by the National Fisheries Service, permit number 514.

## Data Availability

The raw measurements are available in the Supplementary File.

## Supplemental Information

Supplemental information for this article can be found online at http://dx.doi.org/10.7717/peerj.7771#supplemental-information.

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
