# Peer review of "Assessing the effect of fish size on species distribution model performance in southern Chilean rivers"

_PeerJ, doi:10.7717/peerj.7771_

## Round 0.1 · original submission · Major Revisions

The reviewers have commented on your above paper. They indicated that it is not acceptable for publication in its present form. Major modifications are needed regarding implementation of the research comes with too much issues in respect of language, experimental design as well as interpretation and discussion of the results.

Reviewer 1 ·

Basic reporting

The text is well structured however the language needs improvements in several aspects. Throughout the whole text (especially in the discussion) a lot of misspellings, wrong use of adverbs and articles or weird use of singular & plural forms occur, some examples can be found in L97, L241, L283, L304, L338, L340, L351. Furthermore, there are a lot of unclearly formulated sentences that need more clearness. This unconcise formulation leads to a lot of questions for the reader, so it is highly important to clear them out to enable a clear understanding, a good example here is L63 where the authors mention that ‘anthropogenic variables represent the main group of threats to river fishes’ – it is simply unclear what is meant by ‘anthropogenic variables’ and how variables can represent threats. Also, the following conclusion that the role of body size on SDM performance is therefore highly relevant for conservation is simply not following a logical line of argumentation. If there is a specific aspect related to body size that goes beyond the general effect of human-induced pressures, the authors should clearly explain it but this is lacking. This would definitely give additional strength to the manuscript. Another example is in L97, where the size gradient of ‘this species’ is mentioned. Probably the authors mean the size gradient of ‘these species’ which completely changes the meaning of the sentence.
The manuscript only contains 2 tables. However, the authors decided to place the important information about the fish size that was used in the modelling into the supplemental material. (L96). Layout of table 2 can be improved
Data: The authors only provided coordinates of species occurrences. Further evaluation of results was not possible. It is also surprising that the authors used two versions 2.3.3 (L146) and 3.3.2 (L204) of R.

Experimental design

The description of the materials do not provide sufficient detail to appropriately evaluate the quality of the data. Firstly, there is no further information on the quality of the data in the database of the environment ministry that builds an important basis for the analyses. It stays unclear which time period is covered and which methods were used to detect the species. Also, the field sampling is not sufficiently respectively unclearly described. A sampling time of 45-60 minutes seems rather long.
There is no clear justification why the river segments without presences of the fish species of interest can be regarded as true absences. I think it would be much more reliable to use a pseudo-absence approach in this case as probably the information of absences is not available from the ministry database.
Please provide information about the correlations between predictor variables before using them in a GLM method that is sensitive to correlations.
The models are based, in relation to the number of variables, on few observations. Creating a distribution model on 9 observation is inappropriate. Furthermore, model performances are weak (below 0.5) in a lot of cases – see Table 1. These models must be discarded as their predictions are not reliable and not useful for any kind of interpretations. From the results it seems that some species are positively related to human pressures. Hence, doubts are raising that the environmental gradients of the species are fully covered by the data that is used for the modelling. I also have to disagree with the interpretation that the results indicate any impact of anthropogenic uses. Probably, the data mainly covers the human uses and misses the natural environmental gradient but to proof this the amount of data is probably to less.
Please provide information about the size of the area that is covered by the investigation area (estimated from Fig1 it seems rather small). This would be important to judge about the general validity of the results.
The authors did some evaluations in Google Earth. It is not clear how this data was processed to derive land use information. Furthermore, there are global datasets available that describe land cover (e.g. Globcover) and can be also used for the evaluation of land use in Chilean rivers.

Validity of the findings

Data: The authors only provided coordinates of species occurrences. Further evaluation of results was not possible based on the submitted data.
Unfortunately, the discussion represents the weakest part of the manuscript. This section needs a thorough shaping towards the main aim of the manuscript. One example is that the hierarchical framework of habitats sensu Frissell et al, 1986, is firstly mentioned in L312. It makes sense to refer to this concept but this information should be provided to the reader already in the introduction. Another example, in L334ff you somehow contradict yourself by first stating that there is no broad agreement on predictor variables. I would disagree here and somehow you also list then common factors that are used by modellers, such as temperature, discharge, flow velocity. So, I would not say that there is no broad agreement how river ecosystems should be described and several modellers describe the river systems in similar ways but the specific variables they use differ. In a lot of cases this makes sense but of course the direct comparability suffers. However, the biggest issue with this part of the discussion is that it rather stays unclear how the research presented in the manuscript contributes to solve this issue.
A direct comparison of Radinger et al. (2017) and Markovich et al. (2012) is not applicable. The-former is based on field-sampling data, the latter is based on distribution grids so completely different information basis.
There are several issues more in the discussion but as long as the methodogical issues are not solved it may be not useful to elaborate on these issues as the results will look very different after an implementation of the changes.

Additional comments

Although I like the idea of the paper there arise too much issues in the manuscript that have to be solved to provide a solid piece of research. Firstly, try to establish a clear terminology throughout the manuscript in alignment with the recent literature and perform a thorough proof reading of the text. A mixture of misspellings and weak formulations makes it hard to follow your ideas, e.g. in several parts of the manuscript the term ‘turbulence’ is used I assume the authors basically mean flow velocity. I would recommend to fully replace the term turbulence. Furthermore, clearly indicate the limitations of your analyses. You used a small number of occurrences and the quality of the models is not very good in a lot of cases. At some point it is necessary to discard models when there quality drops beneath certain thresholds. Finally, streamline the discussion. You have some statements where the presented research does not really provide any contribution (or you have to better explain how the research contributes)

Reviewer 2 ·

Basic reporting

The English is good in many sections, but needs significant improvement in a few. The statistical methods are very unclear to me, however, which makes evaluation challenging.

The references to the data are generally adequate. However, I think the introduction should make reference to the possibility of differential detection/capture efficiency based on body size as a factor explaining model performance.

Figures and tables look good.

The raw data are included.

Experimental design

The research question is reasonable.

The methods are unclear as written, which makes evaluation difficult.

I think the number of species is really too small to draw any firm conclusions.

Validity of the findings

I think the authors overreach in their interpretations based on the very limited number of species. I think there need to be many more caveats, and much of this is essentially speculative and should be more clearly labeled as such.

Additional comments

The authors test whether fish body size correlates with SDM performance. This is an interesting idea, but I think the sample size (N=7 species) is too small to draw any firm conclusions; indeed, the authors find that depending on the predictor variables employed, the relationship can either be positive or negative. Therefore, I think the authors need to rewrite the abstract, the heading on line 221, and other sections that seem to claim a clear finding that body size and performance are positively related. I think the authors tend to over-interpret some of their other results as well, as noted in the detailed comments.

This is well written in many places, much less so in others (e.g., lines 97-100). There are several places where the singular is used where the plural should be employed, and vice-versa, and some instances of subject-verb agreement (e.g., line 142, should be “model that predicts”).

The statistical methods are very unclear to me as written. They need to be clarified before I can judge whether they are appropriate. For example, was the 10x resampling done for all species to maintain a ~50/50 presence/absence ratio? That’s a lot of work to deal with the prevalence issue, which other authors (e.g., Jimenez-Valverde and Lobo 2006) have shown is not such a problem if handled correctly. It’s also unclear to me how this resampling worked with the 5-fold cross validation and the 30% withholding. See specific comments for further questions.

Introduction, generally: I was surprised to see no mention of detection/capture probability as a factor involved in SDM performance related to body size. I think it’s worth mentioning that species of different sizes may have differential detection probabilities (could be higher or lower, depending on the sampling method).

Detailed comments.
Abstract: I think “overproduction” is meant to be “overprediction.” Right?
Lines 30, 34, etc: I suggest consistency in the use of “SDM” vs “SDMs.” In line 34 it appears that “SDMs” should be used.
37. “detect less food” is confusing to me. I’m not sure what is meant here. If this is all to explain that larger species have larger home ranges, consider just making that point, with a citation.
42. I don’t think “perceives” is the right term. Maybe simply state that the distribution of species with larger ranges tends to be governed by coarser grained predictors.
62. Please add a citation for this imperilment statistic.
97-100. The English needs some major attention here—it looks like this paragraph was skipped in the editing process. E.g., the first clause should read, “These species were selected because they represent…”
125. Briefly define in the text what is meant by “source of flow.”
148-149. 5-fold cross validation implies that 80% of the data were used for fitting. Is this 80% of the 70%? It may be reasonable to do both cross validation and conventional validation, but please explain the logic for doing both.
150-151. Does this sentence refer to the bootstrapping for O. mauleanum? Please clarify.
152. Is this a restatement of 147, or an additional step? This is unclear.
153. How was the consensus calculated? There are multiple methods available for creating a consensus forecast.
181-186. This is confusing to me as written. I’m not sure what the authors have done and why. Is this to secondarily deal with the rebalancing to get the prevalence close to 0.5?
210-212. This confused me. I’m not sure what this is intended to say.
228-230. It’s not clear to me why correlations between predictor variables and body size are being tested. Was this one of the research questions? I’m not sure it should be.
293-309. I think the authors are greatly over-interpreting the results of a very small number of species. I would be very careful about this kind of speculation; in particular, I would cut the paragraph at 303-309.
345-348. The English needs some attention here.

Reviewer 3 ·

Basic reporting

This is an interesting manuscript dealing with species distribution modelling (SDM) and addressing whether SDMs performance or accuracy differ between fish species that have different theoretical maximum length. As presented, I consider that the authors need to improve a number of issues, before this manuscript can be accepted for publication.

For example, it is not clear why understanding the potential impact of body size on SDM performance would be highly relevant for conservation and management planning efforts (lines 64-66), as this study address the theoretical maximum body length, not the individual body length. Also, to my understanding of Table 2, fish species studied have wider distribution than the two watersheds included in this ms., thus it appears that the theoretical relationship that the authors are considering might be biased (if the theoretical maximum body length is considered, then the whole distribution of fish species should be studied).

In any case, the introduction and background described at the beginning of the ms. (lines 28-74) is not suitable nor completely relevant to make a strong case for the research questions studied by the authors. Please add more details at the introduction section to improve the background of research questions, as some concepts are not defined in any section of the ms.: e.g. What is overproduction of biotopes? (line 25); What is a biotope? (line 73); What is variable participation? (line 232). To the knowledge of this reviewer, these concepts are very specific to SDMs, thus they should be explained thoroughly either at the introduction or mat&met.

Figures should be improved. For example, figs 1 and 5 have overlapping labels, fig 4 is confusing (why including all models for all species?). If one model was chosen for each species, it is not necessary to show the results of the other models, unless a proper discussion emerges from it.

Although some raw data (restricted to fish distribution) is supplied, it is impossible to recreate the study as ecological and anthropogenic variables are missing. Please include the raw data as supplementary material.

Finally, I appreciate the authors are not native speakers, but several phrases and paragraphs of the manuscript needed reading twice or three times in order to understand their meaning. I would appreciate if you send the manuscript for edition in order to use clear, unambiguous and professional English language throughout. Unfortunately, most paragraphs of all sections need English edition.

Experimental design

This is an original primary research within the scope of the journal, but the experimental design might be flawed. For example, 1) it is not clear how the authors selected the SDM algorithm for each species; 2) the number of occurrences (calibration vs validation) is not 5-fold (e.g. Brachygalaxias bullocki has 27 occurrences, 30% is 8 occurrences, thus the remaining should be 5x8= 40 occurrences); 3) if there were 10 models for each species, why there is no variability with consensus statistics? (fig 2); 4) correlation is not causality (lines 191-197), thus another proper statistical test should be included.

Probably, some data was not included:
Where is table S1? (line 96)
Where is table S3? (line 230)

Why comparison with prior research was done only considering studies with n>5? (line 207)

According to table 2 (description column) most variables doesn’t appear to be normal. If the authors used Box-Cox procedures to normalise data (line 217), then normality tests should be used, or at least, to analyse q-q plots of data (or histograms).

I suggest to perform a multi-species distribution modelling instead of performing separate SDMs for each species. Although the idea of including the theoretical maximum body length is really interesting, the experimental design appears to be flawed for the reasons indicated at the basic reporting section of this review, but also because correlation is not a proper statistical test to evaluate the effect of body length on TSS. This is a measure of association of two variables, and should not be used to measure the effect of one independent variable (maximum body length) onto another dependent variable (TSS).
If the authors decided to keep SDMs per separate for each species, at least should be included in a revised version:
1) how each model was selected (Eco vs Anthr vs Eco+Anthr)
2) following Allouche et al (2006): overall accuracy, sensitivity and specificity of each model (table 2)
3) another statistical test to evaluate properly the effect of body length on TSS.
4) a measure of goodness of fit for each model (is it really good a model that has TSS values of 0.11, 0.16 or 0.21?)

Validity of the findings

To my understanding, the effect of fish size on SDM performance is not assessed using correlation tests, thus, objective 1 is not fulfilled in this ms. Also, model performance in previous studies was not compared thoroughly, only were selected a number of studies. Further review should be included.

Additional comments

If the authors will stick to taxonomical standards, when mentioning a fish species for the first time (e.g. lines 62-63; 88-91, etc.), please indicate the authority who described them correctly. If the species was changed from one genus to another, the authority should be written between parenthesis, otherwise parenthesis should not be written. Please check all the species included in this study to conform to taxonomical standard.

---

## Round 0.2 · Minor Revisions

Thank you for your work in order to improve your manuscript. There are still some issues to be improved on it. Among them the theoretical maximum length vs individual body length (adding a paragraph dealing with this issue should be sufficient). See the reviewers comments and consider them in a revised version of your manuscript.

Reviewer 2 ·

Basic reporting

Generally good, with some unclear sections. Greatly improved over the previous version.

Experimental design

The design is good.

Validity of the findings

No comment.

Additional comments

This article is much improved from the previous version. I applaud the authors for conscientiously addressing the comments of the reviewers. There are still a few issues to address, but they are relatively minor. Note that in the interest of providing a timely review, I was not as thorough as I normally try to be; in particular, I have not given the discussion a detailed read.

Line numbers refer to the track changes version.
103. This says “spatial distribution models” rather than “species distribution models”, which is more standard and appears in the revised title.
112. Typo here: SMD instead of SDM.
321. “designated the consensus pondering”—this is nonstandard English; I don’t know what it means.
331. Insert here a parenthetical note that this is the true skill statistic or TSS. The term TSS is used later, but is not defined in the revised ms.
368-373. This section is unclear. It says the statistical relationship was assessed, but looking at model performance statistics per se does not constitute a statistical test. Also, don’t say “metrics such as…”; the reader needs to know what was done, not examples of things that were done. Finally, variable importance needs to be defined, as there are different ways to calculate this.
376. “Betareg” sounds like a shorthand term. Is this beta regression?
383-385. I didn’t understand this.
448-449. Here and elsewhere: normality of residuals is just one of the assumptions of regression, and not necessarily the most important one; this is fine to report, but delete the final phrase “thus validating the statistical analyses.”
522-526. This is stylistic, but I suggest leading the discussion with a paragraph interpreting the main findings and discussing their significance, rather than these notes on model performance.
562. “Tend to decrease their specificity”… does this mean “tend to have lower specificity”?
562-564. I’m not following the logic of this.
640. When summarizing a result of this study, don’t cite another paper. It’s not clear why Frissell is cited here.

Reviewer 3 ·

Basic reporting

This is my second time reviewing this manuscript. I wish to thank the authors, as this is a greatly improved version of the ms., especially concerning the English used throughout. However, some of my initial comments were not considered in this version, so I will highlight them when needed.

Regarding figures and tables, I think they can be improved. For example, text on X axis (figure 3) might be changed (species names should be in italics, 0.1. does not need to have period in-between). It seems that this figure is the direct output from an r script, one possibility is to save it as PDF and edit it in Illustrator or other software capable of editing vector format files. Figure 1 should include the percentage of variance explained by the betareg regression. Finally, please use species names consistently, is it Basilichthys microlepidotus or Basilichthys australis the species you are modelling? (check suppl. file 1, the whole manuscript and figures).

In the same figure 3, please indicate what the box-plots are showing: median, inter-quartile range and maximum and minimum? If that is the case, I do not see any difference between the occurrence probability according to predictor variable classes (riparian vegetation and within channel structures) than differences in the median/mean. If there are not statistical differences there is no need to separate each variable in two classes.

Table 1 includes some model performance descriptors, such as TSS, AUC, Acc, Sens and Spec. Although these can be understood rapidly for some readers, I suggest the authors include in mat&met a brief description of each descriptor, indicating (1) what the statistic stands for, (2) how is it calculated, and (3) when they are considered "good". Some of the readers might not be familiar with them and one of the objectives is to assess model performance.

Experimental design

The aim of the authors is to evaluate the relationship between fish size and SDM goodness-of-fit using three approaches:
(1) assess the relationship between fish size and data availability,
(2) assess the relationship between fish size and model performance, and
(3) compare predictor variable participation and patterns according to fish size.

The authors focused on two well-studied southern Chilean river basins, Bueno and Valdivia, and modelled nine native species.

According to the authors objectives, fish size is the key variable to test for. It is important to state at which ecological level the authors are working on, i.e. at the individual level (each individual has its own length measurement, with their own predictors) or at the species level (the whole distribution of every species is used). As presented, it seems that the authors are modelling at the individual level, and part of the discussion is related to individual fish length (lines 293-294), as they mention that introduced species influence fish body size. If introduced species (or any other variable) influence fish body size, is it correct to relate model performance to theoretical fish body length? I think that at least one paragraph dealing with this idea should be included in the discussion section, considering (1) the limits of this approach (theoretical fish body size will remain as theoretical, while contemporary individual fish body size will reflect model performance), (2) caveats and possible errors while using small datasets (see below).

I still don't understand how the authors did the modelling for every species. In the line 159 the authors mention 2- or 3-fold cross validation scheme, e.g. in the case of Diplomystes camposensis they used 14 presences, which are divided in 5 occurrences (for the NNET modelling) and 9 occurrences for the 2-fold cross validation? Is that correct? If that is the case, then proper statistics validating such modelling approach should be included. For example, why not use the MMA data to create the models, and then use the electro-fishing data to validate it? As presented it is not clear why the authors did a field sampling campaign, how many rivers were sampled and how many fish were caught and included in the modelling. Please be more clear.

Another possibility is to use a jackknife approach to validate the model, modelling dozen or hundreds of times each species, leaving 30% of the data for validation.

Regarding statistical analysis and the statistics themselves, please include a full description, not just p-value: lm test (lines 204-25)

Validity of the findings

Shapiro test just indicate if the data (or residuals) are normally distributed, they do not validate statistical analyses (lines 213, 219). Statistical validity should be tested with an independent dataset (such as electro-fishing data), jackkniffe, or using proper statistics (TSS, AUC, etc) or using the confusion-matrix.

To the knowledge of this reviewer Habit et al (2006) is not a field identification manual, it is merely a description of the state of knowledge of fishes in Chile at that time. Please provide a proper identification manual, if it was used.

Additional comments

no comments

---

## Round 0.3 · accepted · Accept

Thank you for improving your manuc¡script according to the provided suggestions and also for submitting your work to this Journal.